# LangMedSAM: A Scalable Adaptation of Medical Segment Anything Model (MedSAM) for Language-Prompted Medical Image Segmentation

## Abstract

Image segmentation is a crucial component of medical imaging, facilitating precise analysis and diagnosis by identifying anomalies and structures across various imaging modalities. Recent advancements have led to the development of foundational medical image segmentation models such as MedSAM. Trained on a large corpus of medical images, MedSAM generates segmentation masks based on user prompts such as bounding boxes and points. For faster inference, LiteMedSAM, a lightweight variant of MedSAM, offers a computationally more practical solution, while maintaining comparable performance. However, manually providing bounding boxes for each 2D slice in volumetric imaging remains cumbersome and hinders the automatic processing of large datasets. To address this, we introduce LangMedSAM, a multi-modal text-based segmentation model that leverages natural language prompts for mask generation in radiological images. LangMedSAM is trained on 20 publicly available medical datasets and evaluated both on these datasets and on 4 additional external datasets to assess generalizability. Building on LiteMedSAM's architecture, it supports segmentation via both text-based prompts and conventional inputs such as bounding boxes. Our results show that text-based prompts provide a scalable and effective solution for multi-modal and multi-region medical image segmentation, offering a practical alternative to conventional prompting methods in MedSAM—particularly for the automated processing of large collections of scans.

## 1 Introduction

Segmentation is a fundamental process in medical imaging that enables precise delineation of anatomical structures, including tissues, organs, and pathological anomalies. This improves diagnostic accuracy, enhances clinical workflows, and supports various research applications. Advances in deep learning have significantly simplified this complex task, with methods such as nnU-Net (Isensee et al., 2018) achieving remarkable accuracy in segmenting regions of interest (ROIs), including various anatomical structures and pathologies. However, most of these algorithms are highly task-specific and exhibit limited generalizability, often performing suboptimally when applied to the segmentation of out-of-domain ROIs. This lack of generalizability has been a persistent issue that poses significant challenges to researchers in the field.

Segment Anything (SAM) (Kirillov et al., 2023) is a foundation model trained on the SA-1B dataset to overcome task-specific limitations of traditional segmentation models in natural images. By leveraging user-provided prompts, such as bounding boxes or points, SAM enables flexible segmentation across diverse tasks, constituting a major advancement in image segmentation techniques. However, medical imaging presents unique challenges, including large image sizes, small and sparse regions of interest, and object classes that are distinguishable only by subtle differences (Xu et al., 2024), resulting in weaker performance than natural images. MedSAM (Ma et al., 2024a) addresses these limitations by fine-tuning SAM on over one million medical image-mask pairs, enhancing segmentation performance across various medical imaging modalities and tasks.

Medical images, especially radiological scans such as Magnetic Resonance (MR) and Computed Tomography (CT), are not a single image but volumetric data composed of multiple 2D slices viewed in axial, coronal, and sagittal planes. Although these slices can be visualized individually as 2D images, segmentation using MedSAM requires input prompts for each slice, since the ROI varies in shape and location across different images. This process becomes impractical, as manually creating bounding boxes or points for each 2D slice is time-consuming and tedious. However, using text prompts instead of conventional prompts can significantly reduce the effort required for segmentation. Text-based prompts allow for automatic segmentation across multiple slices, while images with suboptimal masks can still be manually refined using bounding boxes or points. In addition, text prompts can simultaneously segment multiple ROIs from the same or multiple scans, providing a more scalable solution. Leveraging natural language as a primary mode of interaction, our approach moves beyond purely visual prompting, laying the groundwork for a new generation of flexible and efficient medical image segmentation models.

MedSAM is a large transformer model, and while it provides exceptional segmentation performance, its high computational requirements pose a challenge for experiments in resource-constrained environments. To address this, the developers of MedSAM introduced LiteMedSAM, a lightweight version trained using knowledge distillation (Hinton et al., 2015) with performance comparable to the original model. LiteMedSAM uses a Tiny ViT (Dosovitskiy et al., 2021) as its image encoder, significantly reducing computational overhead.

To move beyond conventional bounding box prompts, we develop LangMedSAM, a model that builds upon LiteMedSAM and incorporates natural language prompts for medical image segmentation. This design not only preserves compatibility with traditional inputs but also unlocks scalable, text-driven segmentation, demonstrating superior flexibility and generalizability across modalities and datasets. Our main contributions are threefold:

1. We introduce LangMedSAM, a new model for medical image segmentation that leverages natural language prompts to directly specify regions of interest, offering a scalable alternative to conventional bounding box–based prompting.

2. Through comparative analysis, we show that LangMedSAM consistently outperforms existing models, even when tested with language prompts of varying phrase lengths, highlighting its robustness to prompt formulation.

3. We study the impact of different text encoders: SAPBERT (Liu et al., 2021), PubMedBERT (Gu et al., 2021), BERT (Devlin et al., 2019) on segmentation performance, and as part of our ablation analysis, we further evaluate LangMedSAM with a contrastive loss mechanism to align image and language embeddings.

## 2 RELATED WORK

Medical image segmentation has long been a focal area of research, attracting significant attention from the scientific community. Numerous models such as nnU-Net (Isensee et al., 2018), SAM-Med2D (Cheng et al., 2023), and MedSAM (Ma et al., 2024a) have been proposed to generate segmentation masks for ROIs. Among these, MedSAM stands out as a versatile and promptable foundational segmentation model. MedSAM fine-tunes Meta's Segment Anything Model (SAM) (Kirillov et al., 2023) on medical image datasets, achieving state-of-the-art results. Furthermore, these foundational models have been integrated with open-source DICOM viewers to enhance accessibility and facilitate a wider adoption in medical imaging applications (Yildiz et al., 2024a;b; Ma et al., 2024b). However, it relies exclusively on bounding boxes and point-based prompts, making manual annotation across multiple images a tedious task. This highlights the need to incorporate text-based prompting into segmentation models to improve efficiency and scalability.

Recent research has explored the use of text-based prompts for object detection and segmentation in both natural and medical images. Grounding DINO (Liu et al., 2024) detects objects in natural images by combining localization losses such as L1 and GIoU (Rezatofighi et al., 2019) with a contrastive loss that aligns predicted objects with language embeddings. While effective in natural image domains, its generalization to medical images is limited due to differences in texture, scale, and complexity. To address this gap, SimTxtSeg (Xie et al., 2024) adapts textual cues for medical images by generating pseudo-bounding boxes with a text-to-visual converter, refining them with

SAM, and training a hybrid text–vision attention decoder, achieving promising results in polyp and brain tumor segmentation. Complementary to these approaches, CRIS (CLIP-Driven Referring Image Segmentation) (Wang et al., 2022) leverages contrastive learning to align textual features with pixel-level representations, enabling segmentation based on natural language queries; however, its applicability to medical images remains unexplored, motivating our ablation study to assess its effectiveness in this domain. More recently, MedCLIP-SAM (Koleilat et al., 2024) introduces another design pathway by deriving bounding boxes from saliency maps refined with a Conditional Random Field (Kraehenbuehl & Koltun, 2013), which are then used as prompts for MedSAM to perform segmentation.

However, approaches that rely on text prompts to generate bounding boxes face notable limitations when used for segmentation. Bounding boxes often cover overly large regions, especially when the regions of interest (ROIs) are spatially separated, leading to the inclusion of unintended areas. They also struggle to capture fine anatomical details, such as thin-walled structures like the myocardium (see Figure 1). Point-based prompts provide more localized guidance, but they require precise placement of both positive and negative points—a process that is labor intensive and time-consuming, ultimately reducing annotation efficiency.

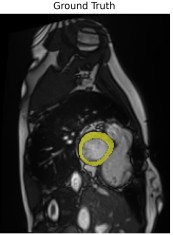 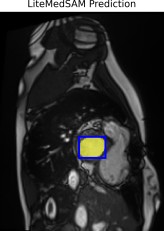

Figure 1: Example of imprecise segmentation using bounding box prompting: the bounding box provided for the myocardium leads to segmentation of the entire heart region, highlighting the limitations of spatial prompts in isolating fine-grained anatomical structures.

Another recent advancement is BiomedParse (Zhao et al., 2024), a segmentation model developed by Microsoft that supports text-based prompts. Trained on over three million image-mask-text triplets, BiomedParse leverages GPT-4 to align unstructured textual information with established biomedical object ontologies. However, BiomedParse is limited to textual inputs, and architecturally lacks support for visual prompts such as bounding boxes or point-based annotations. Additionally, Biomed-Parse is computationally expensive, requiring 16 GB of VRAM during inference, which limits its usability across different platforms and its integration with open-source DICOM viewers. In contrast, we introduce LangMedSAM, a lightweight segmentation model that supports both natural language and visual prompts, providing flexible interaction modes for medical image analysis. Its efficiency allows seamless deployment on widely available hardware and straightforward integration into existing medical software and DICOM viewers, offering a more practical alternative to large-scale models such as BiomedParse.

## 3 METHODOLOGY

Current SAM-based methods in the medical domain rely on bounding boxes or point-based prompts, both of which limit scalability due to annotation effort and precision requirements. To overcome these limitations, we extend LiteMedSAM into a text-driven framework—LangMedSAM—that unifies language and vision, enabling segmentation directly from textual descriptions. The model consists of three key components: an image encoder, a prompt encoder, and a mask decoder.

The image encoder $E_{img}$ processes an input image $I^{H \times W \times C}$ to produce a corresponding image embeddings $I_{emb} \in \mathbb{R}^{H/4 \times W/4 \times D}$:

$$I_{emb} = E_{img}(I^{H \times W \times C}) \tag{1}$$

Here $H, W$ and $C$ are the height, width and number of channels of the input image $I$, respectively. $D$ refers to the output dimension of the image encoder. We introduce a multimodal prompt encoder

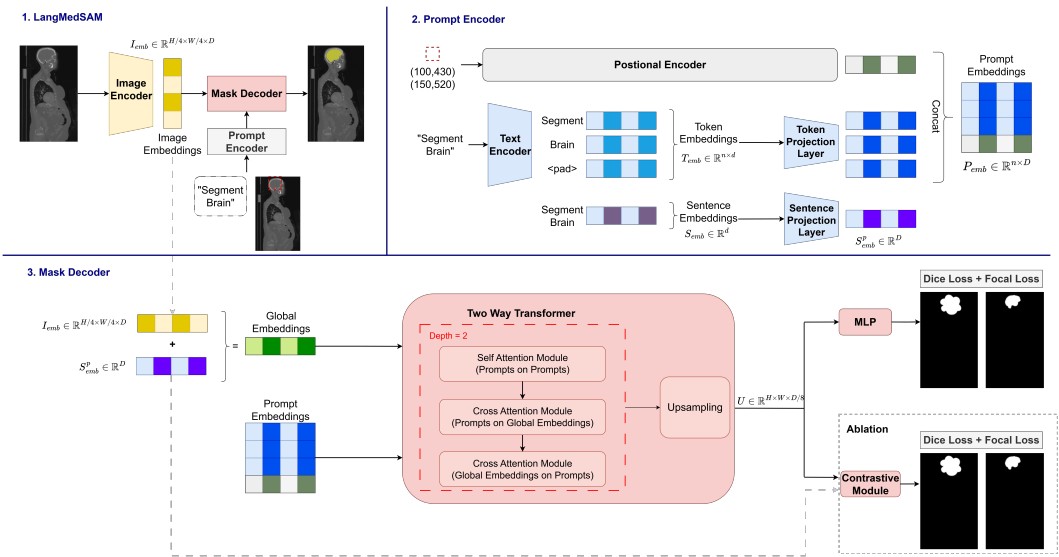

Figure 2: Overview of LangMedSAM - Block 1 presents the overall architecture of LangMedSAM. Block 2 details the incorporation of text prompts within the prompt encoder. Block 3 illustrates the mask decoder for text-guided segmentation

that extends LiteMedSAM beyond spatial inputs, incorporating text embeddings from pre-trained text encoders such as BERT, PubMedBERT, or SAPBERT. Unlike prior SAM-based methods restricted to bounding boxes or points, this architecture enables segmentation to be guided directly by natural language prompts ($P_{text}$). The text encoder ($E_{text}$) produces a global sentence-level representation ($S_{emb} \in \mathbb{R}^d$), which captures the overall semantics of the prompt and serves as a high-level guidance signal for segmentation. Beyond this global embedding, $E_{text}$ also generates token-level embeddings that preserve the granularity of individual words. By incorporating both sentence- and token-level information, LangMedSAM can attend not only to the overall intent of a prompt but also to fine-grained cues such as explicit class names or anatomical terms. This dual-level design enables robust segmentation even when prompts are long (50 words or longer), as the model can still reliably identify and localize the relevant structures if the target class appears anywhere within the description. To accommodate prompts of varying lengths, we adopt the standard padding mechanism used in transformer-based encoders (e.g., BERT). This ensures that token-level embeddings are produced in a consistent format across the batch. For a sentence with $n$ tokens (including padding), the token embeddings are denoted as $T_{emb} \in \mathbb{R}^{n \times d}$ where $d$ is again the output dimension of text encoder. Thus, the encoding process can be summarized as:

$$S_{emb}, T_{emb} = E_{text}(P_{text}) \tag{2}$$

To ensure compatibility with the image encoder and mask decoder, the sentence and token embeddings are each passed through dedicated linear projection layers, referred to as the sentence projection and token projection layers, respectively. These are followed by a GELU activation to introduce non-linearity and preserve expressive capacity:

$$S_{emb}^p = \phi_{GELU}(MLP_{proj}^{sent}(S_{emb})) \tag{3}$$

$$T_{emb}^p = \phi_{GELU}(MLP_{proj}^{tok}(T_{emb})) \tag{4}$$

Here, $MLP$ denotes a multilayer perceptron, $\phi_{GELU}$ denotes the GELU activation function, $S_{emb}^p \in \mathbb{R}^D$ and $T_{emb}^p \in \mathbb{R}^{n \times D}$ are projected sentence and token embeddings, respectively. $D$ represents the hidden dimensionality used by the transformer layers in the mask decoder, which matches that of the image encoder. The transformed token embeddings are concatenated with any additional prompts (such as bounding boxes), if provided, to form prompt embeddings:

$$P_{emb} = Concat(T_{emb}^p, PositionalEnc(Box, Points)) \tag{5}$$

Here, $P_{emb} \in \mathbb{R}^{n \times D}$ denotes the prompt embeddings. Our experiments are exclusively designed to evaluate the effectiveness of text-based prompting; therefore, no other prompt types are employed.

Consequently, the number of tokens produced by the text encoder directly corresponds to the number of tokens in the prompt embedding.

Finally, the image, sentence, and prompt embeddings are jointly processed by the mask decoder, which has been adapted to fuse both visual and textual cues. While the underlying design draws inspiration from LiteMedSAM, our modifications extend its capabilities to language-driven representations, enabling flexible multimodal segmentation. Similar to other text-based segmentation and detection models (Xie et al., 2024; Liu et al., 2024), our mask decoder builds on a two-way transformer architecture with multi-head attention mechanisms, including both self-attention and cross-attention. Unlike prior works, we extend this design by incorporating explicit attention masking (Vaswani et al., 2023), which not only handles padded tokens but also allows selective control over information flow between text and visual features. This modification is particularly important in our setting, as prompts may consist of long natural language descriptions, and masking ensures that irrelevant or padded tokens do not interfere with the fine-grained alignment of language and image features. As a result, the decoder also receives self and cross attention masks $M_S$ & $M_C$ where $M_{C/S}^{ij} \in \{0, -\infty\}$. The $-\infty$ value masks the embedding of i$^{th}$ token from attending to the embedding of j$^{th}$ token in the prompt, to ensure that padding tokens do not influence the attention computations.

The two-way transformer itself comprises three key modules: the first module performs masked multi-head self-attention (MHSA) among various types of prompts and tokens, enabling the model to capture inter-dependencies between them:

$$P_{emb}^1 = P_{emb} + LayerNorm(MHSA(Q = K = V = P_{emb}, M_S)) \tag{6}$$

The second module conducts multi-head cross-attention (MHCA) between the computed prompt embeddings and the image embeddings, which are augmented with sentence embeddings to form global embeddings, allowing the model to learn how prompts and tokens attend to visual and textual features:

$$P_{emb}^2 = LayerNorm(P_{emb}^1 + MHCA(Q = P_{emb}^1, K = I_{emb} + S_{emb}^p, V = I_{emb}, M_C)) \tag{7}$$

Conversely, the third module applies multi-head cross-attention in the opposite direction—from the combined image and sentence embeddings to the prompts—enabling the model to understand how image and textual information influence the prompts:

$$I_{emb}^1 = LayerNorm(I_{emb} + MHCA(Q = I_{emb} + S_{emb}^p, K = P_{emb}^2, V = P_{emb}^2, M_C)) \tag{8}$$

Here, $I_{emb}^1 \in \mathbb{R}^{H/4 \times W/4 \times D}$. While the mask decoder can be stacked with multiple two-way transformer blocks, we adopt the two-block configuration from LiteMedSAM, which provides a strong balance between segmentation accuracy and computational efficiency. These transformer blocks fuse visual and textual features into enriched image embeddings that capture both spatial structure and semantic context. The fused embeddings are then passed through a lightweight upsampling module, where transposed convolution layers progressively restore spatial resolution and reduce channel dimensionality:

$$U = Upscale(I_{emb}^1) \tag{9}$$

where, $U \in \mathbb{R}^{H \times W \times D/8}$. Finally, a multi-layer perceptron (MLP) is applied to the upsampled embeddings to adjust the channel dimensions and generate the binary segmentation mask.

As an additional experiment, we explore a text-to-pixel contrastive learning strategy, inspired by CRIS (Wang et al., 2022), to examine whether explicit alignment between text embeddings and pixel-level predictions can further improve segmentation. To this end, we introduce a contrastive module (Figure 2) that projects the sentence embedding into the image embedding space and learns weights and biases to guide alignment:

$$W_t = MLP(S_{emb}^p) \quad ; \quad B_t = MLP(S_{emb}^p) \tag{10}$$

Here, $(W_t \in \mathbb{R}^{D/8 \times 3 \times 3})$ and $B_t \in \mathbb{R}$ are weights and biases respectively. In essence, this process transforms the sentence embeddings into a $3 \times 3$ convolutional kernel, which is then applied across the upscaled feature map to produce a final segmentation mask.

$$Mask = (U \circledast W_t) + B_t \tag{11}$$

where $\circledast$ denotes the convolution operation. This effectively allows for the sentence embeddings to interact with each pixel in the upscaled embedding. Instead of the binary cross-entropy (BCE) loss originally used in CRIS (Wang et al., 2022), we adopt focal loss (Lin et al., 2020), which has demonstrated success in segmentation tasks, while still serving the objective of aligning textual and visual representations. For both contrastive and non-contrastive models, focal loss is combined with dice loss to more effectively guide the training process.

## 4 DATASET AND EXPERIMENTS

To train LangMedSAM, we utilize 20 publicly available radiological datasets (mentioned in Table 1) encompassing two imaging modalities: magnetic resonance (MR) and computed tomography (CT). From each dataset, we extract representative 2D slices to construct the training set, ensuring coverage of diverse anatomical regions and pathologies. CT images have intensity values ranging from -2000 to 2000, while MR images range from 0 to 3000. To standardize intensity values, CT images are normalized using typical window width and level settings. Most CT datasets use a window width of 400 and a level of 40. However, specific anatomies and pathologies require tailored contrast settings. For example, the HaN-Seg (Podobnik et al., 2024) dataset, focused on head and neck regions, uses a window width of 1500 and a level of -500. Datasets targeting lung abnormalities, such as MSD-LungTumor (Antonelli et al., 2022) and COVID-19-CT-Seg (Ma et al., 2021), apply a width of 1500 and a level of -600. For hepatic vessel segmentation in the MSD-Hepatic dataset (Antonelli et al., 2022), the window width and level are set to 300 and 120, respectively. For MR images, intensity values are clipped between the 0.5th and 99.5th percentiles. Both CT and MR images are then rescaled to the $[0, 255]$ range. To satisfy the dimensionality requirement of the image encoder, all images were resized to a fixed dimension of $256 \times 256 \times 3$. As CT and MR images typically contain only one channel, we duplicate the single channel three times to match the required input format.

Table 1: List of datasets used for model training. Datasets marked with * were used exclusively as part of the external test set

| | |
|---|---|
| TotalSegmentator (Wasserthal et al., 2023) | ACDC (Bernard et al., 2018) |
| CHAOS (Kavur et al., 2021) | MSD-Heart (Antonelli et al., 2022) |
| MSD-Prostate (Antonelli et al., 2022) | HaN-Seg (Podobnik et al., 2024) |
| PROMISE (Litjens et al., 2014) | QUBIQ (Becker et al., 2019) |
| AMOS (Ji et al., 2022) | MSD-Brain (Antonelli et al., 2022) |
| MSD-Lung (Antonelli et al., 2022) | MSD-Pancreas (Antonelli et al., 2022) |
| MSD-Hepatic Vessel (Antonelli et al., 2022) | MSD-Colon (Antonelli et al., 2022) |
| ATLAS Bourgogne (Quinton et al., 2023) | Covid-19-CT Seg (Cohen et al., 2020) |
| CrossMoDA (Dorent et al., 2023) | KiTS23 (Heller et al., 2023) |
| EMIDEC (Lalande et al., 2020) | MRBrainS18 (Kuijf et al., 2024) |
| M&Ms* (Campello et al., 2021) | BTCV* (Gibson et al., 2018) |
| SegTHOR* (Lambert et al., 2019) | WORD* (Luo et al., 2022) |

To ensure effective training, we exclude images with very small masks (fewer than 100 pixels) and those containing multiple disjoint masks for the same structure. For text prompts, we use a diverse set of phrasings, such as "Extract the {class name} from the image", "Highlight and extract {class name} from the scan" and "Separate {class name} from surrounding structures" among others (see Appendix Table 6). These prompts are processed through the text encoder, which tokenizes them and computes the corresponding text embeddings. The maximum length of each text prompt is limited to 256 tokens. Our training dataset contains 180 unique classes (see Appendix Table 5). In total, 1 million image-text-mask triplets are selected, with 750,000 used for training and 250,000 for validation. The model is trained on a single H200 GPU of 144 GB memory, till convergence using the AdamW optimizer ($\beta_1 = 0.9$ and $\beta_2 = 0.999$), with an initial learning rate of 1e-4 and a weight decay of 0.01. The batch size is set to 32. We train the entire pipeline while keeping the pre-trained text encoder frozen. During inference, LangMedSAM requires only 700 MB of VRAM to segment the ROI. Further information about the inference speed of different models is mentioned in Appendix Table 8.

We evaluate LangMedSAM against state-of-the-art segmentation models, including MedSAM and its lighter variant LiteMedSAM. Unlike prior works that rely on text-to-box generation or pseudo-

mask supervision, we exclude such approaches from our main comparison. Because MedSAM is explicitly trained on bounding boxes, it naturally serves as an upper bound for evaluation metrics and thus provides a more meaningful benchmark than methods that derive bounding boxes indirectly from text. To ensure consistency and rigor, we construct two large-scale evaluation sets: (i) an internal set with 200,000 unseen 2D scans from training datasets and (ii) an external set with 20,000 unseen 2D scans from the external datasets, where all models are assessed under identical conditions.

Beyond bounding-box–based models, we further compare LangMedSAM with BiomedParse, a segmentation model that directly leverages text prompts. For this comparison, we assemble a benchmark of 15,000 2D slices drawn from diverse datasets (ACDC, AMOS, MSD-Brain, and KiTS23), covering a wide spectrum of anatomies and pathologies including myocardium, liver, aorta, brain tumors, and kidney tumors. Both LangMedSAM and BiomedParse are evaluated with identical prompts of varying lengths, where 1–3 word prompts correspond to class names, and longer prompts include five sentence variants per class to assess consistency across phrasing styles. For evaluation, we report the mean dice similarity coefficient (DSC) and normalized surface dice (NSD) to assess pixel-wise overlap and boundary alignment between predicted and ground truth masks. Additional details on the datasets has been provided in the Appendix material (see Appendix Table 4).

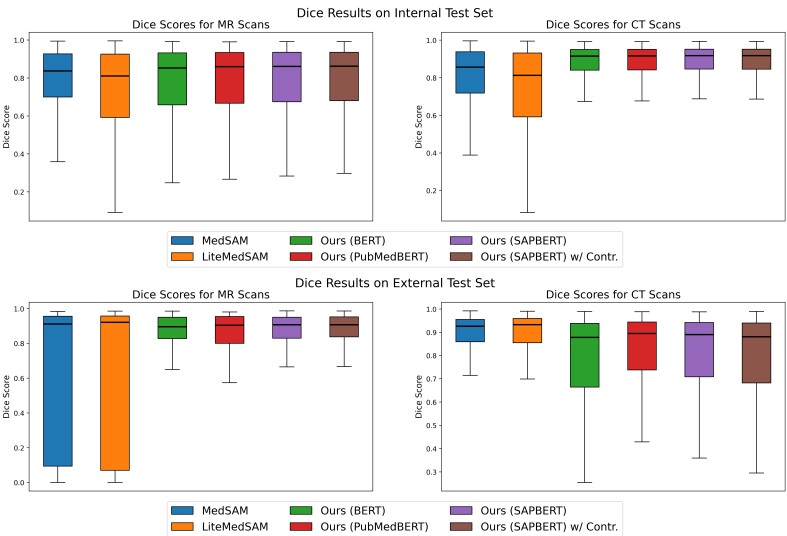

Figure 3: Dice Score Comparison on Internal and External Test Sets: Each plot presents the Dice scores achieved by different models across different datasets. The left subplot shows performance on MR scans, while the right subplot shows performance on CT scans. Results are reported separately for the internal test set (top plot)which was held-out from training datasets, and the external test set (bottom plot), comprising entirely unseen datasets. In each plot, the center line within each box indicates the median, while the lower and upper edges of the box represent the 25th and 75th percentiles, respectively. "w/ Contr." refers to model trained with text-to-pixel contrastive learning, respectively.

Table 2: Performance comparison of LangMedSAM and baselines in terms of Dice similarity coefficient (DSC) and normalized surface dice (NSD, 2 mm tolerance) on internal and external test sets for MR and CT modalities.

| Models | Internal Dataset | | | | External Dataset | | | |
|---|---|---|---|---|---|---|---|---|
| | MR | | CT | | MR | | CT | |
| | DSC | NSD | DSC | NSD | DSC | NSD | DSC | NSD |
| MedSAM | 0.77 | 0.58 | 0.79 | 0.61 | 0.63 | 0.76 | 0.88 | 0.68 |
| LiteMedSAM | 0.71 | 0.55 | 0.71 | 0.55 | 0.63 | 0.77 | 0.86 | 0.70 |
| Ours (BERT) | 0.73 | 0.57 | 0.85 | 0.74 | 0.86 | 0.82 | 0.74 | 0.56 |
| Ours (PubMedBERT) | 0.74 | 0.58 | 0.85 | 0.74 | 0.86 | 0.82 | 0.77 | 0.59 |
| Ours (SAPBERT) | 0.74 | 0.59 | 0.85 | 0.75 | 0.86 | 0.82 | 0.77 | 0.59 |
| Ours (SAPBERT) w/ Contr. | 0.75 | 0.59 | 0.85 | 0.75 | 0.86 | 0.83 | 0.75 | 0.57 |

Table 3: Dice similarity coefficients for LangMedSAM and BiomedParse evaluated with equivalent text prompts of different lengths.

| Model | Word count in text prompts | | | | | | | |
| --- | --- | --- | --- | --- | --- | --- | --- | --- |
| | 1∼3 | | 10∼12 | | 25∼30 | | 50∼60 | |
| | MR | CT | MR | CT | MR | CT | MR | CT |
| Ours(SAPBERT) | 0.72 | 0.88 | 0.72 | 0.88 | 0.72 | 0.88 | 0.71 | 0.87 |
| Ours (PubMedBERT) | 0.72 | 0.88 | 0.72 | 0.88 | 0.72 | 0.88 | 0.69 | 0.82 |
| Ours (BERT) | 0.72 | 0.88 | 0.72 | 0.88 | 0.72 | 0.88 | 0.71 | 0.87 |
| BiomedParse | 0.52 | 0.81 | 0.51 | 0.67 | 0.48 | 0.35 | 0.47 | 0.33 |

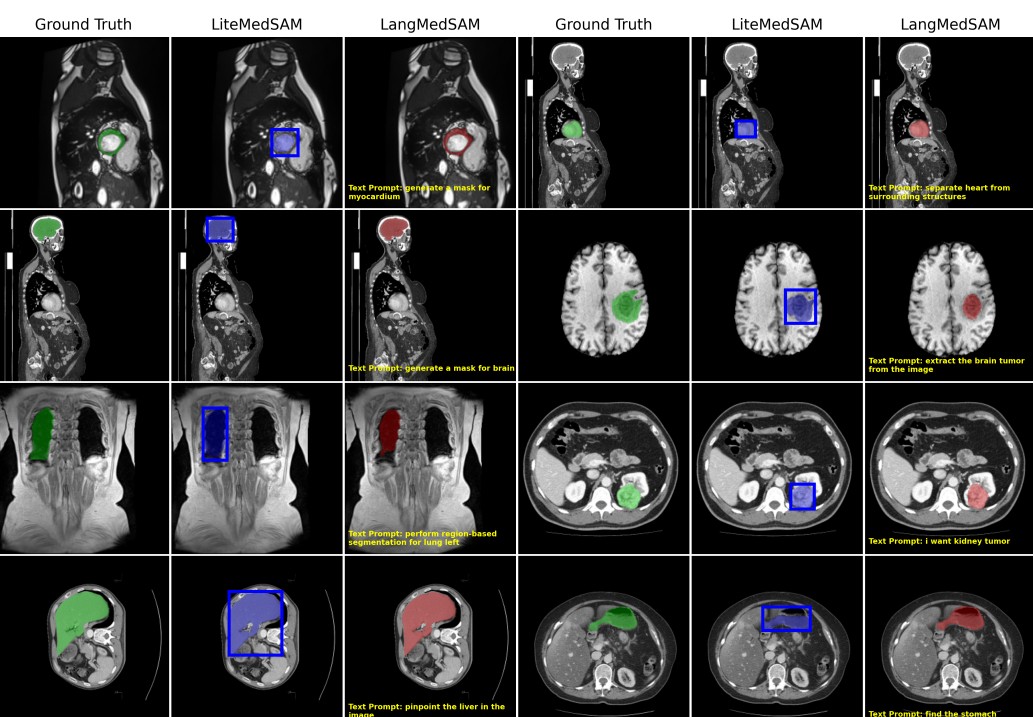

Figure 4: Qualitative comparison of segmentation results. Each row displays a set of samples arranged in a grid of 6 columns. Columns 1 and 4 show the original images with ground truth masks overlaid. Columns 2 and 5 present the segmentation outputs from LiteMedSAM, where bounding box prompts are provided. Columns 3 and 6 illustrate the results from LangMedSAM (SAPBERT) using natural language prompts, with relevant text highlighted. The first three rows correspond to examples from the internal test set, while the final row showcases samples from the external test set.

## 5 RESULTS AND ANALYSIS

We evaluated different versions of LangMedSAM incorporating textual prompts and compared them with the baseline MedSAM and LiteMedSAM models, both of which rely on visual prompts—specifically, oracle bounding boxes derived directly from segmentation masks. Figure 3 (top plot) provides a visual comparison of model performance across MR and CT modalities on the internal test set. As expected, LiteMedSAM demonstrates lower performance compared to its larger counterpart, MedSAM, reflecting the performance trade-offs inherent in knowledge distillation. In contrast, all LangMedSAM variants perform competitively on MR scans, achieving mean dice scores between 0.73 and 0.75. The performance of different models on internal test set are also tabulated in Table 2. It shows that each language-prompted model outperforms the visually prompted LiteMedSAM, which averages at a dice score of 0.71. The normalized surface dice (NSD) at a 2mm tolerance is also consistent across language-prompted models and slightly exceeds that of LiteMedSAM. On CT scans, the advantage of language prompts becomes even more pronounced.

LangMedSAM variants achieve a dice score of 0.85 compared to 0.71 from LiteMedSAM—a relative improvement of approximately 20%. This trend is mirrored in NSD values as well.

The true robustness of language-prompted segmentation models is tested on external datasets. As shown in Table 1, these include M&Ms, BTCV, SegTHOR, and WORD, of which only M&Ms offers MR scans. Model performance on the external test set is presented in Figure 3, with additional metrics provided in Table 2. The M&Ms dataset presents additional challenges due to its irregularly shaped myocardium structures (illustrated in Figure 1), leading to a broader dice score distribution for MedSAM and LiteMedSAM on MR images, as seen in Figure 3 (bottom plot). In contrast, LangMedSAM performs better by effectively learning the myocardium appearance from training data. The CT scans in the external test set span a diverse range of thoracic and abdominal organs, posing generalization challenges. Consequently, language-prompted models show a drop in performance relative to visual models. For instance, LiteMedSAM achieves a Dice score of 0.86, compared to 0.77 for LangMedSAM (SAPBERT). This discrepancy may stem from the inherent strengths of visual prompts, which offer direct spatial cues, whereas textual prompts require models to infer spatial regions from language hindering precise localization. This difficulty is also reflected through reduced NSD scores for language-guided models on external CT scans compared to their visually prompted counterparts. Although visual prompts retain an advantage in certain settings, the consistently strong Dice scores achieved by LangMedSAM demonstrate that language-driven prompting can provide a powerful and scalable pathway toward generalized medical image segmentation.

We further investigated the impact of different text encoders, experimenting with general-purpose BERT and its medically specialized variants, SAPBERT and PubMedBERT. Our results suggest that both medical-domain models offer slight advantages over standard BERT, particularly evident in CT scans from the external test set (Table 2), where they achieve marginally higher dice and NSD scores. Moreover, models using BERT tend to exhibit broader dice score distribution (as seen in Figure 3, bottom-right plot for CT), indicating less consistent performance compared to their domain-adapted counterparts. We also compared contrastive and non-contrastive training paradigms. As illustrated visually in Figure 3 and quantitatively in Table 2, contrastive learning did not yield noticeable improvements. We hypothesize that the multi-head self- and cross-attention mechanisms used to fuse text and image features are already sufficiently powerful for effective segmentation, making additional contrastive alignment redundant in this setting.

Finally, we also evaluated LangMedSAM against BiomedParse by providing both models with comparable text prompts of varying lengths. As shown in Table 3, LangMedSAM maintains consistently high segmentation performance across different prompt styles and complexities, demonstrating strong robustness to input variation. In contrast, BiomedParse exhibits a marked decline in CT segmentation accuracy as prompt length increases. We attribute LangMedSAM's stability to its dual use of token-level and global sentence embeddings, which together capture both fine-grained details and broader contextual cues—enabling reliable mask generation even with long or complex textual descriptions.

## 6 CONCLUSION

We introduced LangMedSAM, a lightweight text-driven segmentation model that extends beyond visual prompts to directly leverage natural language in medical image segmentation. Our experiments demonstrate that text prompts can achieve competitive accuracy while offering clear benefits in scalability and usability. By enabling entire scans to be segmented from text prompts—without manual slice-level annotations—LangMedSAM improves efficiency and also supports data annotation. Moreover, its speed and lightweight design allow seamless integration with DICOM viewers and clinical software, reducing barriers to deployment. By uniting linguistic reasoning with visual understanding, this work highlights the potential of language-driven interaction as a foundation for more flexible medical AI systems. Looking ahead, we aim to extend LangMedSAM within MedSAM-2 (Ma et al., 2024b), enabling seamless multi-modal prompting with text, bounding boxes, and points for improved segmentation of complex 3D volumetric scans. Together, these contributions open a promising path toward medical foundation models that combine efficiency, flexibility, and clinical usability. We will release the code for LangMedSAM after reviews to support further research and reproducibility.

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

## A  APPENDIX

In the following appendix, we present supplementary material to support our main findings. This includes a brief note on LLM usage during manuscript preparation, dataset details and accessibility, definitions of evaluation metrics, and additional experimental results.

### A.1  LLM USAGE

In preparing this manuscript, we used publicly available large language models (ChatGPT, DeepSeek) as general-purpose writing assistants. Their role was limited to refining grammar, improving readability, and rephrasing text for clarity. No part of the research design, data analysis, experiments, or results relied on LLMs.

### A.2  DATASET DETAILS AND ACCESSIBILITY

LangMedSAM was trained on 20 publicly available medical datasets and evaluated on 4 additional external datasets to assess its generalizability. A comprehensive list of these datasets is provided in Appendix Table 4. Together, they span a broad spectrum of anatomical structures and pathological conditions. In total, the dataset encompasses 180 unique classes, detailed in Appendix Table 5.

Table 4: Overview of the datasets used for training and evaluation. Each entry includes the imaging modality, number of scans, target segmentation regions, and a link to the corresponding dataset website. Datasets marked with * were used exclusively for testing.

| Dataset | Modality (No. of Scans) | Target Segmentation (Modality) | Link |
|---|---|---|---|
| TotalSegmentator | CT (1228), MR (616) | 117 different classes (CT), 50 different classes (MR) | TotalSegmentator |
| ACDC | MR (150) | Left Ventricle Cavity, Right Ventricle Cavity, Myocardium | ACDC |
| CHAOS | CT (30), MR (40) | Liver (CT, MR), Kidneys (MR), Spleen (MR) | CHAOS |
| HaN-Seg | CT (42) | Head and Neck Organs | HaN-Seg |
| PROMISE | MR (50) | Prostate | PROMISE |
| QUBIQ | CT (30), MR (50) | Pancreas (CT), Pancreatic Lesion (CT), Brain tumor (MR), Prostate(MR) | QUBIQ |
| AMOS | CT (500), MR (100) | 15 Abdominal Organs | AMOS |
| ATLAS Bourgogne | MR (60) | Liver, Liver Tumor | Atlas Bourgogne |
| Covid-19-CT Seg | CT (20) | Covid Infection | Covid-19 |
| CrossMoDA | MR (105) | Vestibular Schwannoma Tumor | CrossMoDA |
| KiTS23 | CT (489) | Kidneys, Kidney Tumor | KITS23 |
| EMIDEC | MR (100) | Myocardium, Infarcted Myocardium, No-Reflow Area | EMIDEC |
| MRBrainS18 | MR (30) | Brain regions | MRBrainS18 |
| MSD-Heart | MR (20) | Heart | MSD |
| MSD-Prostate | MR (48) | Prostate | MSD |
| MSD-Brain | MR (484) | Brain Tumor | MSD |
| MSD-Lung | CT (96) | Lung Tumor | MSD |
| MSD-Pancreas | CT (282) | Pancreas, Pancreatic Tumor | MSD |
| MSD-Hepatic Vessel | CT (303) | Hepatic Vessels, Hepatic Tumor | MSD |
| MSD-Colon | CT (126) | Colon Cancer | MSD |
| M&Ms* | MR (25) | Left Ventricle, Right Ventricle, Myocardium | M&M |
| BTCV* | CT (30) | 14 Abdominal organs | BTCV |
| SegTHOR* | CT (40) | Heart, Aorta | SegThor |
| WORD* | CT (20) | 15 Abdominal Organs | WORD |

Table 6 outlines the diverse set of text prompts used during LangMedSAM training. Owing to this variety, the model learned to focus on key class names within prompts. As a result, it can effectively extract the target regions even when presented with novel phrasings—such as "I want {text}"—that were not explicitly seen during training. Additionally, Appendix Table 7 presents the number of samples included in the training, validation, test, and external test sets for each modality, MR and CT, respectively. Further, Appendix Table 8 provides comparison between the speeds of all the models that are compared. The results show that while LangMedSAM (SAPBERT) is slightly heavier than LiteMedSAM, it is still much faster and lighter than BiomedParse. Although we could not precisely compute GFLOPS for BiomedParse due to its modular architecture, its significantly higher inference time and more complex computation graph strongly suggest a substantially higher computational cost compared to both LiteMedSAM and LangMedSAM.

### A.3  EVALUATION METRICS

To evaluate and compare the performance of the proposed LangMedSAM with models like LiteMedSAM, MedSAM and BiomedParse, we use two metrics: Dice Similarity Coefficient (DSC), Normalized Surface Dice (NSD).

Table 5: List of 180 anatomical and pathological classes used for medical image segmentation. These classes span multiple organ systems and include both healthy structures and disease-related entities, covering a diverse range of anatomical regions for comprehensive segmentation tasks.

| | | | |
|---|---|---|---|
| myocardium | left ventricle cavity | right ventricle cavity | right kidney |
| left kidney | buccalmucosa | glottis | parotid right |
| lips | eye posterior right | glnd submand left | glnd thyroid |
| cavity oral | glnd submand right | esophagus s | eye posterior left |
| larynx sg | spinalcord | bone mandible | parotid left |
| brainstem | cricopharyngeus | glnd leftacrimal right | opticnrv left |
| eye anterior left | artery carotid right | opticnrv right | pituitary |
| artery carotid left | opticchiasm | eye anterior right | cochlea left |
| glnd leftacrimal left | cochlea right | arytenoid | prostate |
| brain tumor | kidney | pancreas | pancreatic lesion |
| bladder | postcava | arota | duodenum |
| liver | gall bladder | stomach | spleen |
| left adrenal gland | right adrenal gland | esophagus | left atrium |
| prostate peripheral zone | prostate transitional zone | lung tumor | pancreatic tumor |
| hepatic tumor | hepatic vessel | colon cancer | liver without tumor |
| liver tumor | covid infection | vestibular schwannoma tumor | kidneys |
| kidney tumor | infarcted myocardium | no-reflow area | basal ganglia |
| cerebrospinal fluid in the extracerebral space | white matter | cortical gray matter | lung left |
| spinal cord | aorta | inferior vena cava | vertebrae |
| lung right | intervertebral discs | heart | colon |
| hip left | autochthon left | iliopsoas right | kidney right |
| gluteus maximus left | gluteus medius left | sacrum | hip right |
| iliopsoas left | kidney left | iliac vena right | gluteus maximus right |
| gluteus minimus left | iliac vena left | iliac artery right | gluteus medius right |
| small bowel | gallbladder | autochthon right | portal vein and splenic vein |
| scapula left | clavicula left | humerus left | clavicula right |
| humerus right | scapula right | brain | adrenal gland right |
| femur left | femur right | gluteus minimus right | urinary bladder |
| adrenal gland left | iliac artery left | lung upper lobe left | lung upper lobe right |
| lung lower lobe left | lung middle lobe right | vertebrae L3 | vertebrae T11 |
| lung lower lobe right | vertebrae T12 | vertebrae L5 | sternum |
| vertebrae L1 | skull | vertebrae L4 | vertebrae T8 |
| vertebrae S1 | vertebrae T10 | vertebrae L2 | vertebrae T9 |
| trachea | vertebrae T1 | rib right 7 | vertebrae T6 |
| vertebrae C2 | rib right 2 | vertebrae T3 | costal cartilages |
| vertebrae T7 | superior vena cava | rib left 5 | brachiocephalic vein left |
| vertebrae T5 | rib right 6 | rib left 8 | rib right 11 |
| vertebrae C4 | rib right 10 | vertebrae C7 | kidney cyst left |
| rib left 10 | rib right 1 | rib left 7 | vertebrae C5 |
| vertebrae T2 | vertebrae T4 | atrial appendage left | rib left 6 |
| rib right 5 | rib left 9 | vertebrae C3 | rib left 4 |
| rib right 4 | kidney cyst right | thyroid gland | rib right 8 |
| brachiocephalic vein right | vertebrae C1 | vertebrae C6 | rib right 9 |
| subclavian artery right | common carotid artery left | pulmonary vein | rib left 1 |
| rib right 3 | subclavian artery left | rib left 2 | rib left 3 |

Table 6: Examples of text prompts used during the training of the LangMedSAM model. In each prompt, the placeholder 'text' is substituted with one of the 180 anatomical segmentation classes.

| | | |
|---|---|---|
| Extract the {text} from the image | {text} | Isolate {text} |
| Mark {text} | Identify {text} | Segment {text} from the image |
| Segment {text} | Extract {text} | Highlight the {text} in the image |
| Locate the {text} | Find the {text} | Detect {text} |
| Pinpoint the {text} in the image | Outline the {text} | Show the {text} region |
| Where is the {text}? | Point out the {text} | Focus on the {text} |
| Crop around the {text} | Annotate the {text} | Separate {text} from the rest of the image |
| Identify and segment the {text} | Perform segmentation on {text} | Delimit the boundaries of {text} |
| Highlight and extract {text} from the scan | Find and mark {text} in the given image | Automatically segment {text} |
| Draw contours around the {text} | Classify and segment {text} within the image | Generate a mask for {text} |
| Extract the region corresponding to {text} | Perform region-based segmentation for {text} | Separate {text} from surrounding structures |
| Generate segmentation output for {text} | Locate and outline {text} | Perform instance segmentation for {text} |
| Highlight and classify {text} in the scan | | |

Table 7: Number of samples in the training, validation, test, and external test sets for Magnetic Resonance (MR) and Computed Tomography (CT) modalities.

| | MR | CT |
|---|---|---|
| Training set | 177,868 | 572,132 |
| Validation set | 43,177 | 206,823 |
| Test set | 45,748 | 154,252 |
| External test set | 414 | 19,386 |

The Dice Similarity Coefficient (DSC) quantifies the pixel-wise overlap between the predicted segmentation and the ground truth mask. It is defined as:

$$\text{DSC} = \frac{2|G \cap P|}{|G| + |P|} \tag{12}$$

Table 8: Detailed comparison of GFLOPS and inference times for all models, measured on an RTX A6000 GPU. Inference times were averaged over 100 runs for consistency.

| Model | GFLOPS | Inference Time (ms/image) |
|---|---|---|
| MedSAM | 488 | 107.85 |
| LiteMedSAM | 40 | 17.34 |
| BiomedParse | - | 4301.88 |
| Ours (SAPBERT) | 70 | 22.47 |

where $P$ is the set of predicted pixels and $G$ is the set of ground truth pixels.

The Normalized Surface Dice (NSD) evaluates how closely the boundaries of the predicted mask align with the ground truth, within a defined tolerance. For our experiments, the tolerance was fixed at 2mm. NSD is defined as:

$$\text{NSD} = \frac{|\{x \in \partial P \mid \exists y \in \partial G, \|x - y\| < \tau\}| + |\{y \in \partial G \mid \exists x \in \partial P, \|y - x\| < \tau\}|}{|\partial P| + |\partial G|} \quad (13)$$

where $\partial P$ and $\partial G$ are surface (boundary) points of predicted mask and ground truth mask respectively. $\tau$ refers to the tolerance threshold. The numerator counts the number of surface points that are within the tolerance distance from the other mask's surface. Whereas, the denominator is the total number of surface points from both masks.

## A.4 FURTHER RESULTS

As noted earlier, our model supports both text and bounding box prompts. After completing the main experiments with text inputs, we further trained the model using randomly mixed prompts (text, boxes, or both). Appendix Table 9 reports the results on the external test set when evaluated with bounding box or text prompts.

Table 9: Performance of LangMedSAM on the external test set when prompted with bounding boxes or text inputs.

| Model | DSC | | NSD | |
|---|---|---|---|---|
| | MR | CT | MR | CT |
| LangMedSAM (SAPBERT) Prompts: Bounding Box | 0.80 | 0.88 | 0.80 | 0.69 |
| LangMedSAM (SAPBERT) Prompts: Text | 0.86 | 0.74 | 0.83 | 0.74 |

To facilitate detailed comparison, we plot the dice scores for each anatomical region and pathology across different models for internal (Appendix Figures 6 - 11) and external test sets (Appendix Figure 5). Appendix Figure 12 showcases segmentation results from LangMedSAM (SAPBERT) on a single image using different textual prompts that were not seen during training. We also present qualitative results comparing LiteMedSAM, which uses oracle bounding boxes, with LangMedSAM, which relies on textual prompts. For fair comparison, the corresponding ground truth images and masks are also included. Sample results from the internal test set are shown in Figs.13 and 14, while Figs.15–17 display results from the external test set.

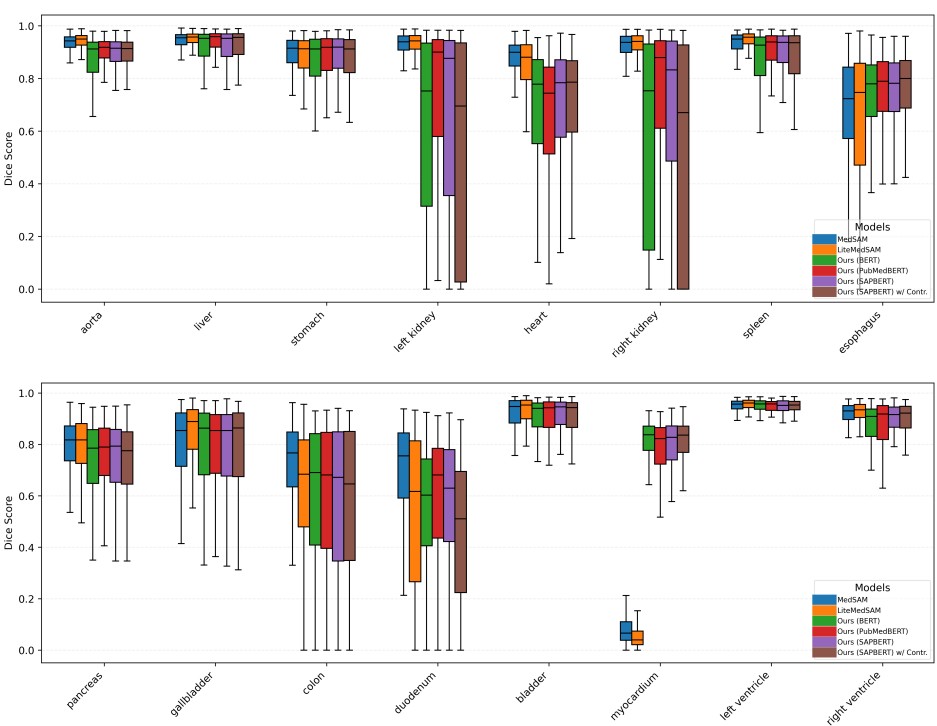

Figure 5: Box plots of Dice scores for each anatomical structure and pathology across all six models on the external test set. Each box plot shows the distribution of Dice scores, where the horizontal line indicates the median, and the lower and upper edges represent the 25th and 75th percentiles, respectively.

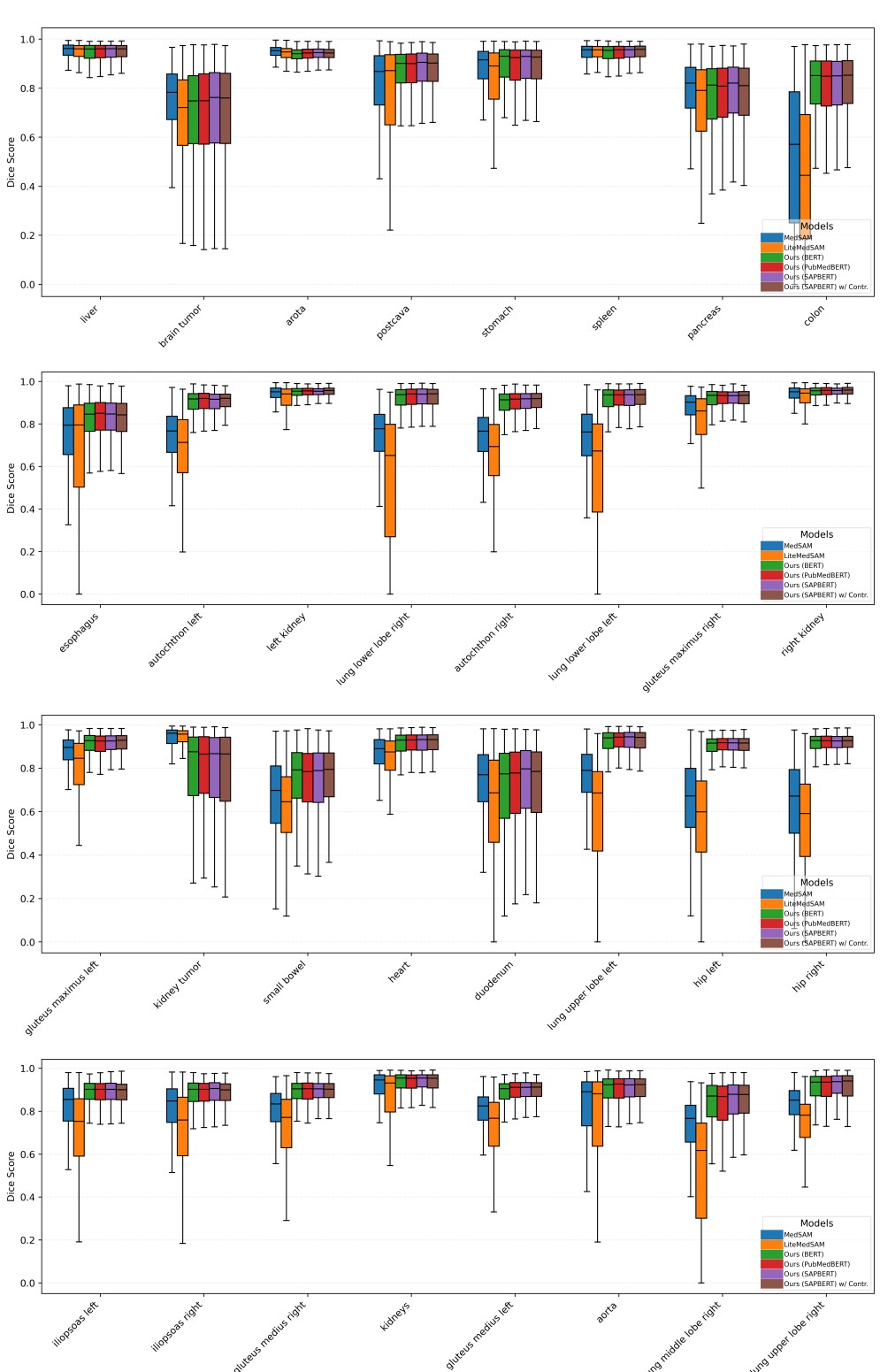

Figure 6: Box plots of Dice scores for each anatomical structure and pathology across all six models on the internal test set. Each box plot shows the distribution of Dice scores, where the horizontal line indicates the median, and the lower and upper edges represent the 25th and 75th percentiles, respectively.

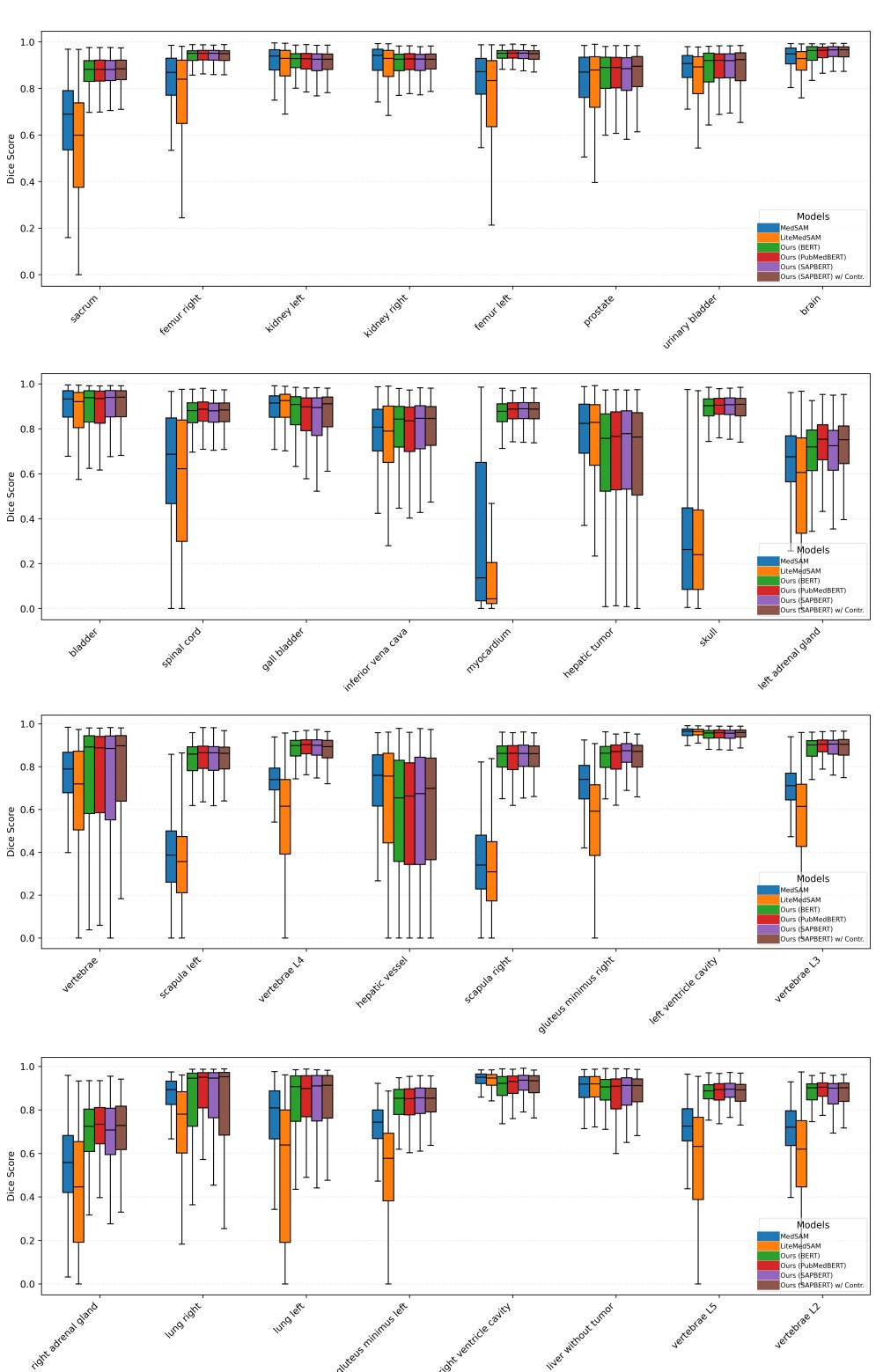

Figure 7: Box plots of Dice scores for each anatomical structure and pathology across all six models on the internal test set. Each box plot shows the distribution of Dice scores, where the horizontal line indicates the median, and the lower and upper edges represent the 25th and 75th percentiles, respectively.

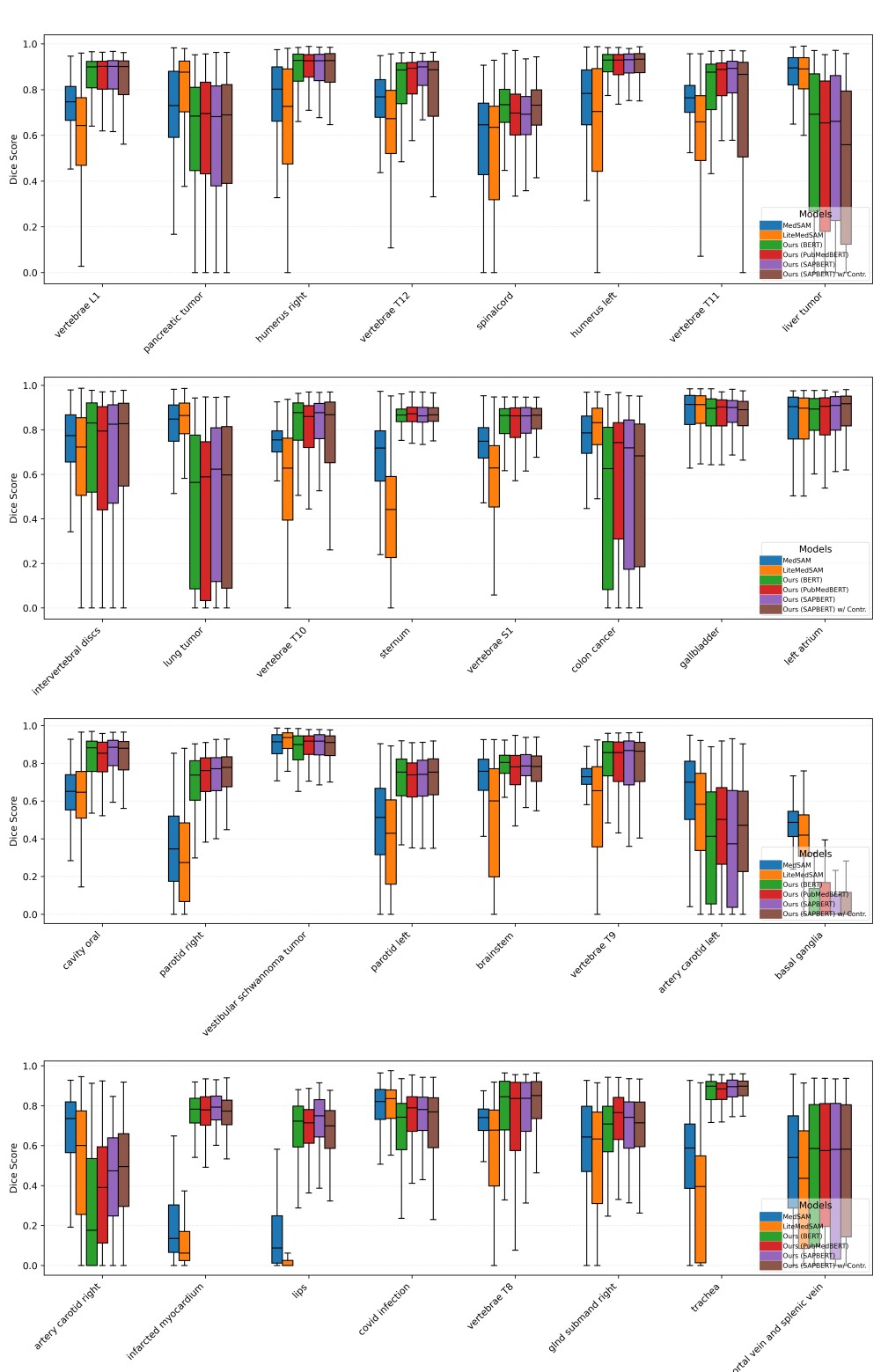

Figure 8: Box plots of Dice scores for each anatomical structure and pathology across all six models on the internal test set. Each box plot shows the distribution of Dice scores, where the horizontal line indicates the median, and the lower and upper edges represent the 25th and 75th percentiles, respectively.

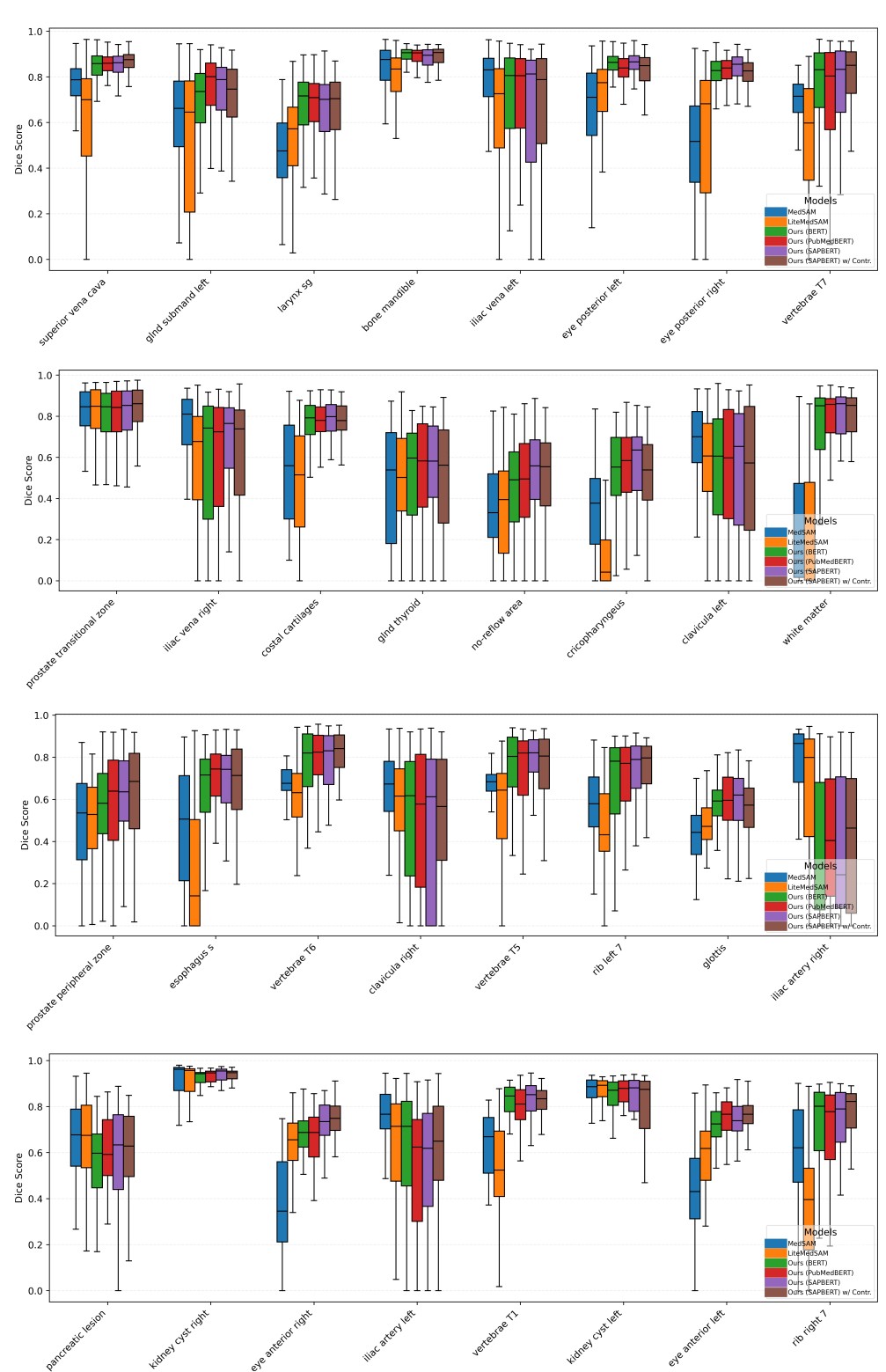

Figure 9: Box plots of Dice scores for each anatomical structure and pathology across all six models on the internal test set. Each box plot shows the distribution of Dice scores, where the horizontal line indicates the median, and the lower and upper edges represent the 25th and 75th percentiles, respectively.

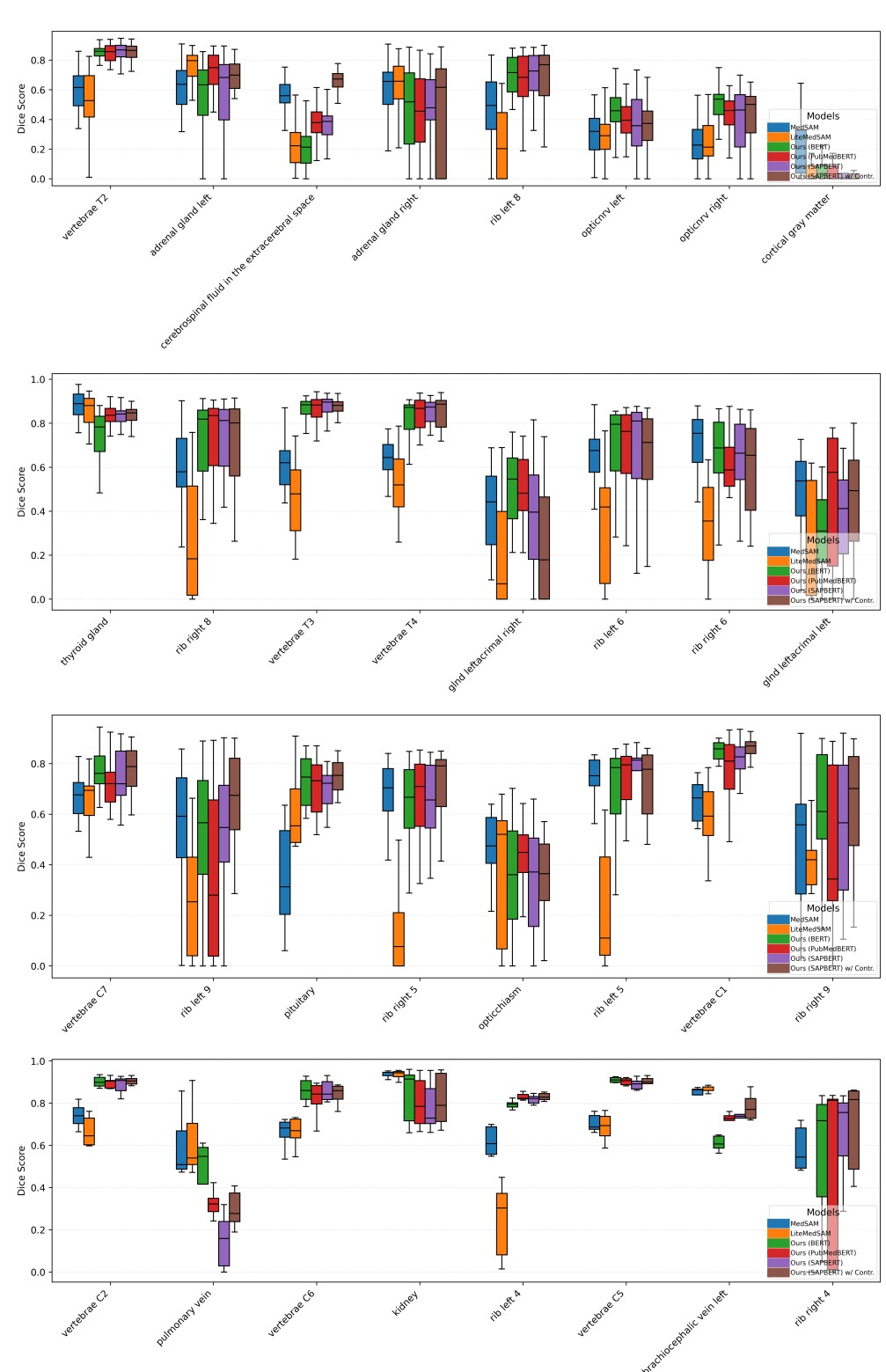

Figure 10: Box plots of Dice scores for each anatomical structure and pathology across all six models on the internal test set. Each box plot shows the distribution of Dice scores, where the horizontal line indicates the median, and the lower and upper edges represent the 25th and 75th percentiles, respectively.

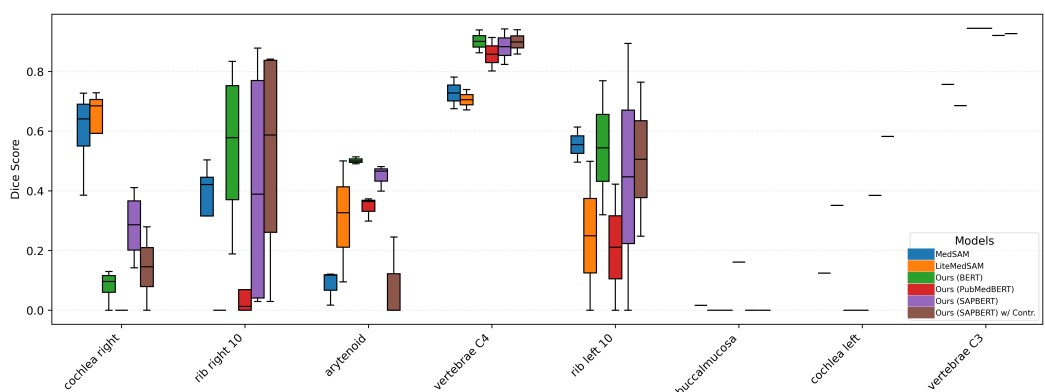

Figure 11: Box plots of Dice scores for each anatomical structure and pathology across all six models on the internal test set. Each box plot shows the distribution of Dice scores, where the horizontal line indicates the median, and the lower and upper edges represent the 25th and 75th percentiles, respectively.

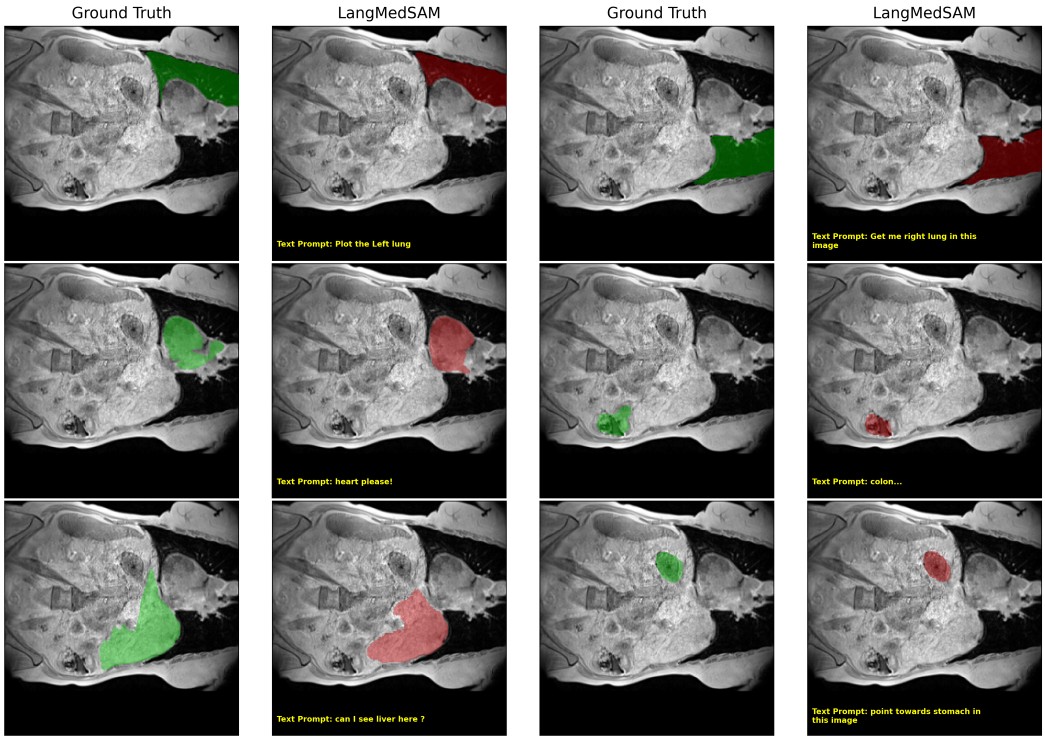

Figure 12: Segmentation results from LangMedSAM (SAPBERT) on a single MR image from internal test set using different textual prompts (unseen during training). Ground truth masks are shown for comparison. The results demonstrate the model's ability to distinguish between anatomical structures and pathologies based on the provided text prompts.

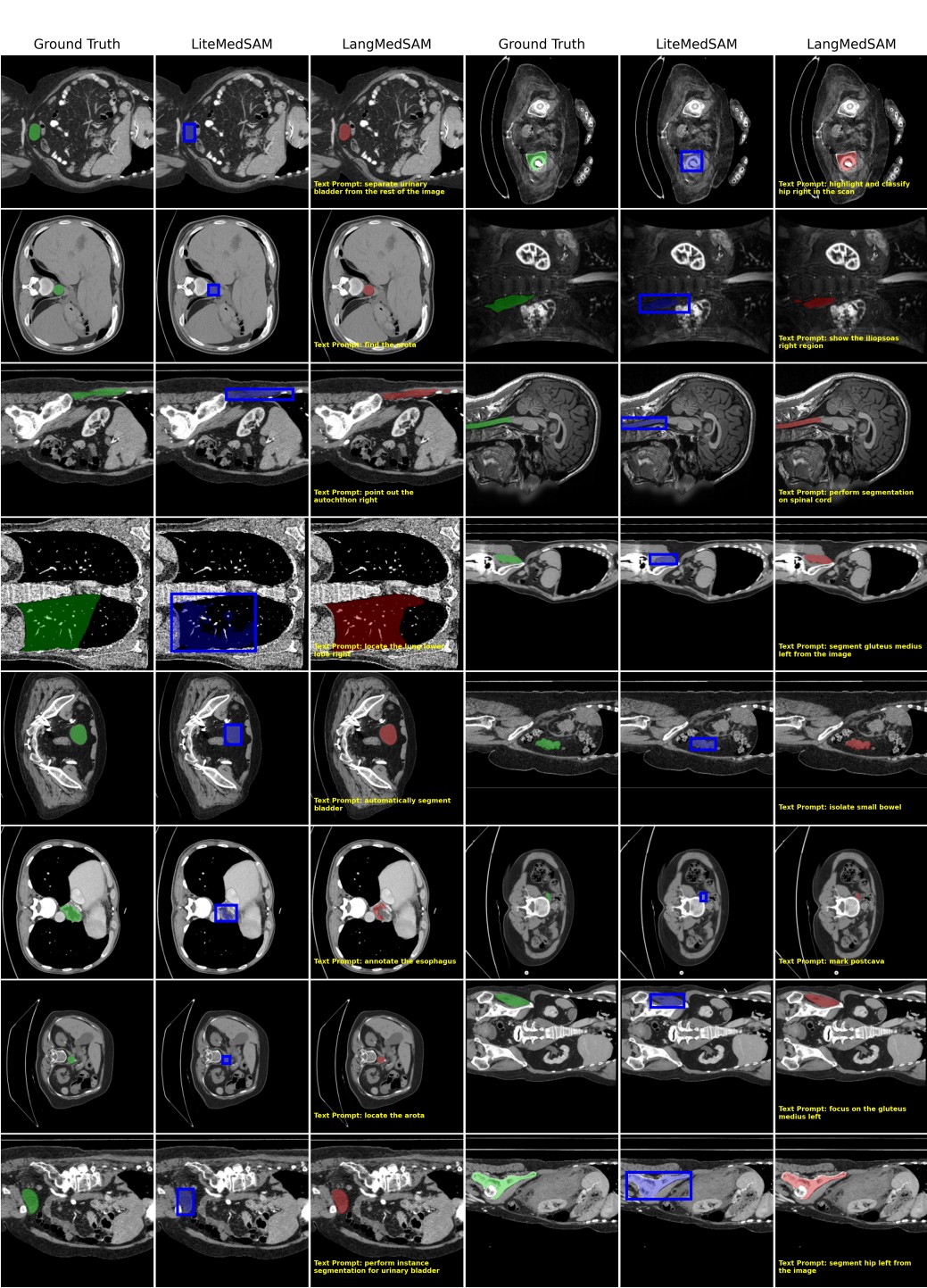

Figure 13: Qualitative comparison of segmentation results on the internal test set between LiteMed-SAM (using oracle bounding boxes) and LangMedSAM (SAPBERT).

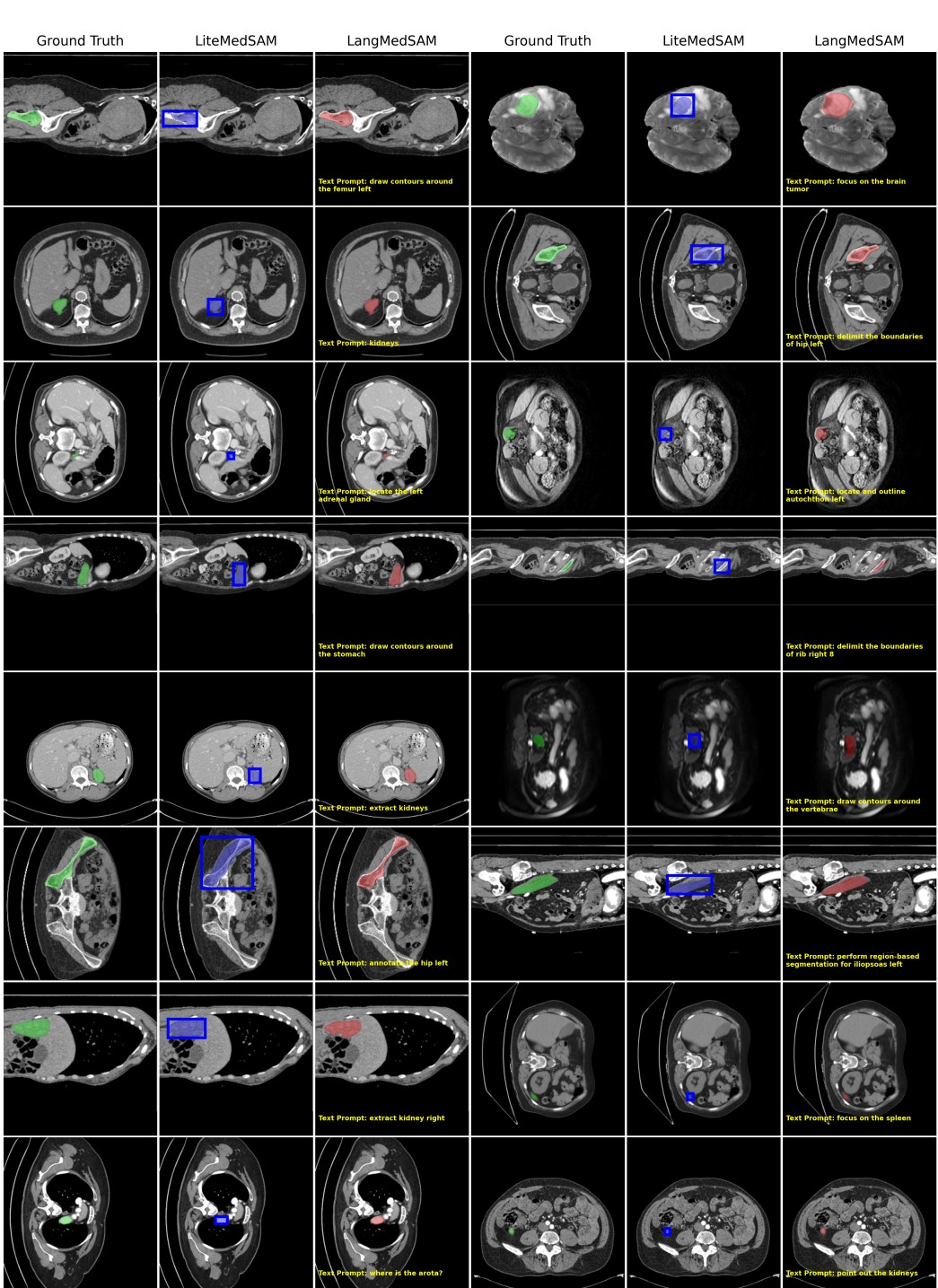

Figure 14: Qualitative comparison of segmentation results on the internal test set between LiteMed-SAM (using oracle bounding boxes) and LangMedSAM (SAPBERT).

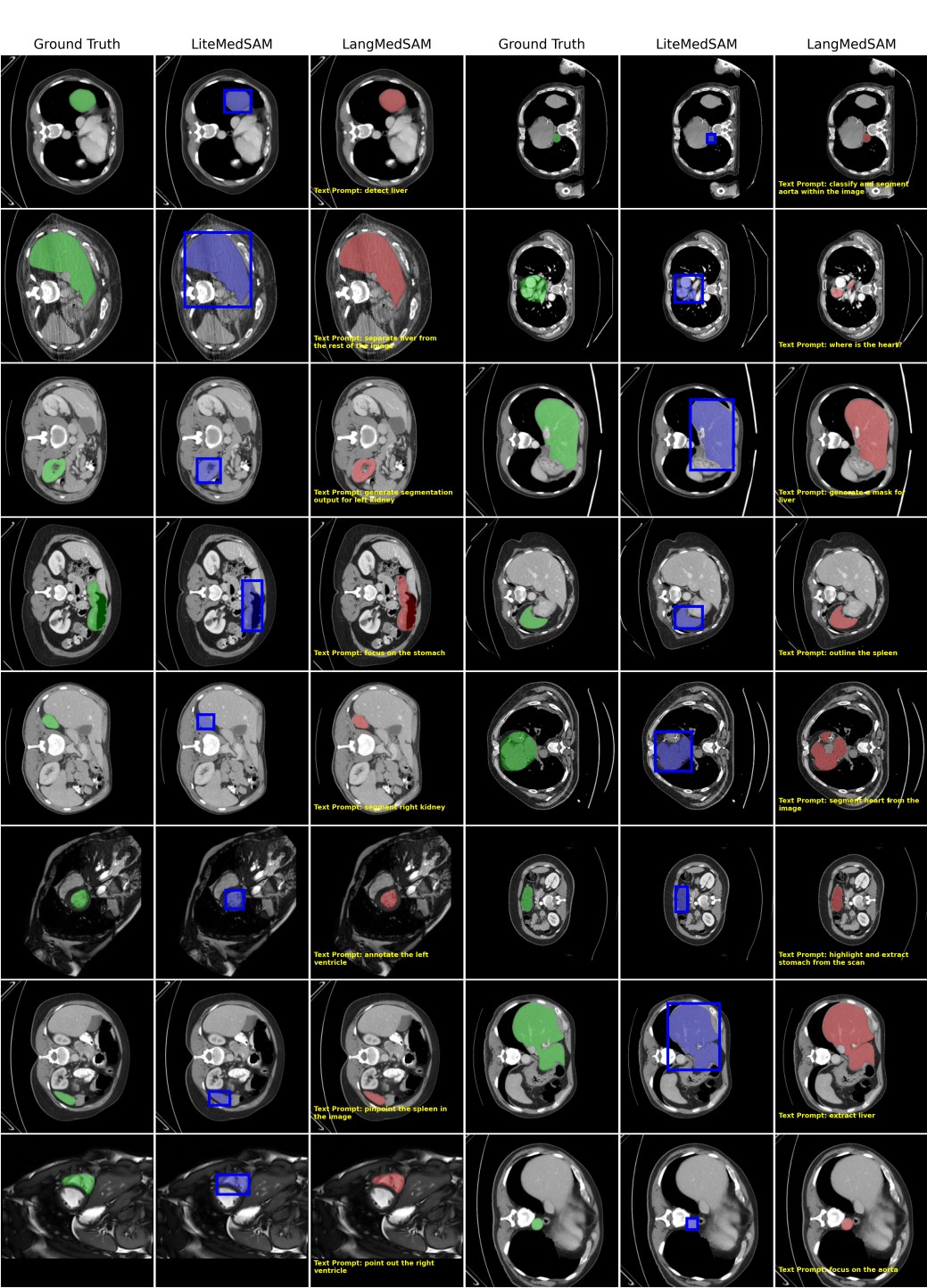

Figure 15: Qualitative comparison of segmentation results on the external test set between LiteMed-SAM (using oracle bounding boxes) and LangMedSAM (SAPBERT).

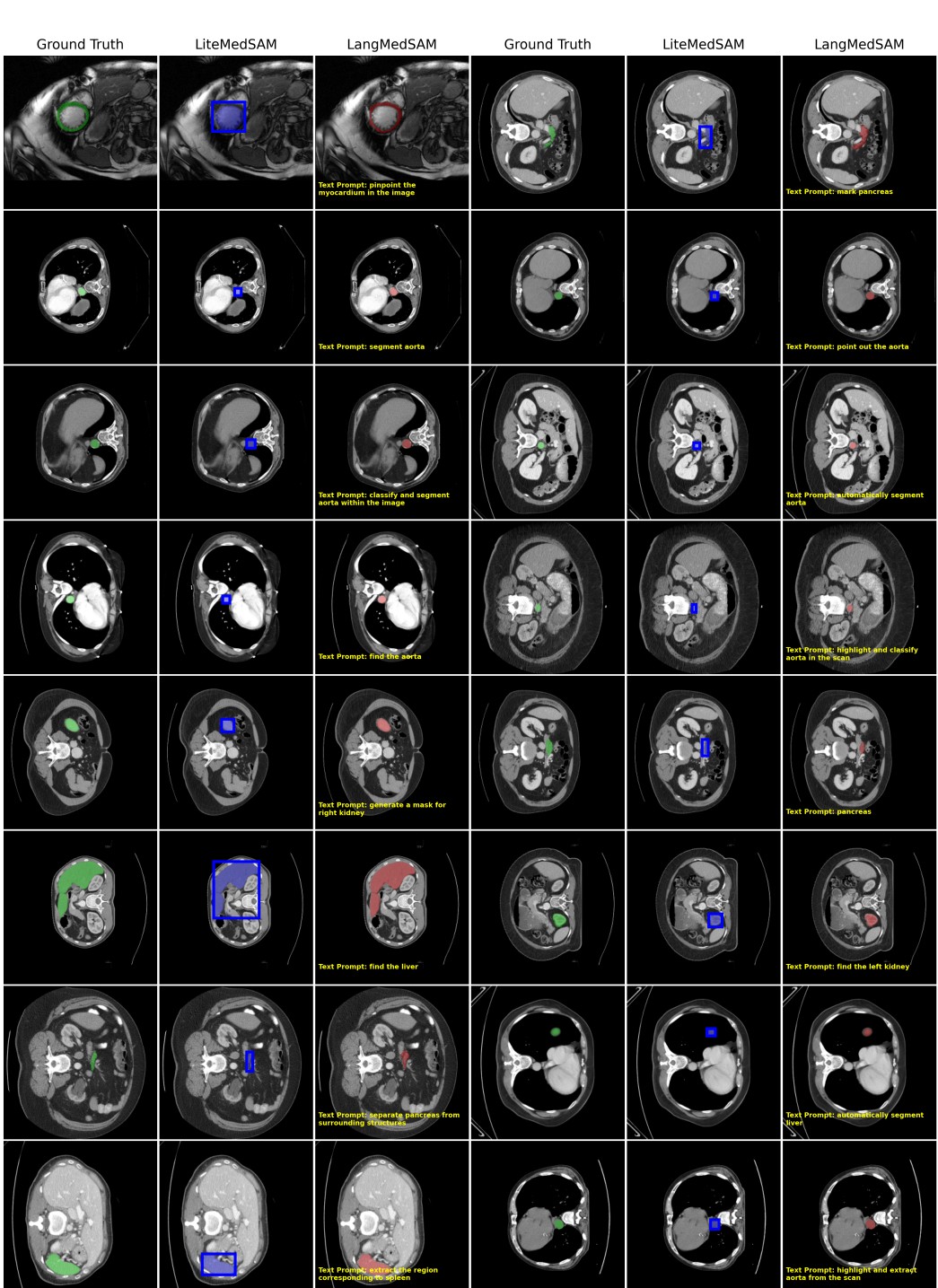

Figure 16: Qualitative comparison of segmentation results on the external test set between LiteMed-SAM (using oracle bounding boxes) and LangMedSAM (SAPBERT).

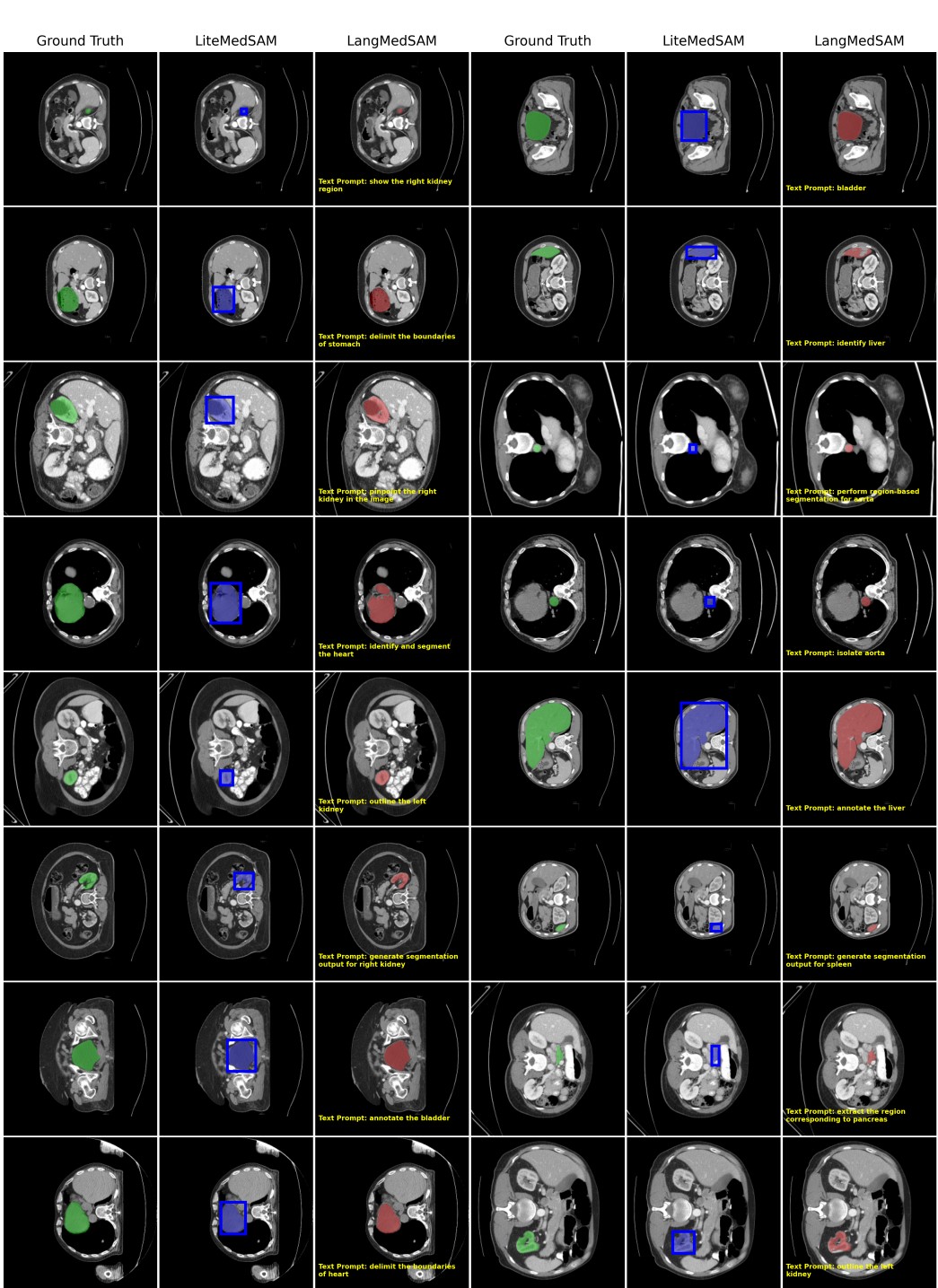

Figure 17: Qualitative comparison of segmentation results on the external test set between LiteMed-SAM (using oracle bounding boxes) and LangMedSAM (SAPBERT).

