# OpenReview forum: "LangMedSAM: Scalable Adaptation of Medical Segment Anything Model (MedSAM) for Language-Prompted Medical Image Segmentation"
_ICLR.cc/2026/Conference — ICLR 2026 Conference Withdrawn Submission_

### Official Review · Reviewer_MuQb · 2025-10-28

**Soundness:** 3
**Presentation:** 2
**Contribution:** 2
**Rating:** 4
**Confidence:** 4

**Summary:**

This paper proposes LangMedSAM, a language-prompt segmentation model for medical imaging, aiming to address the issue that traditional SAM-like models rely on slice-by-slice bounding box prompts in medical scenarios, which leads to high annotation costs. While retaining the lightweight structure of MedSAM, this method introduces a language encoding module to realize the fusion of natural language and visual prompts, thereby enabling direct segmentation of target regions through text descriptions.

**Strengths:**

- It supports hybrid text and visual prompts and is compatible with existing models in the SAM family.

- It conducts thorough comparisons with baseline models, including MedSAM, LiteMedSAM, and BiomedParse.

- It has low inference memory consumption, enabling easier deployment in clinical systems.

**Weaknesses:**

- Although LangMedSAM structurally integrates text encoding and visual prompt modules, its overall framework still follows that of SAM and its derivatives. It is essentially an engineering optimization work, with insufficient innovation and no breakthroughs in theoretical methods or training paradigms.

- In the general computer vision field, there are already multiple segmentation models supporting text prompts (e.g., Grounded-SAM, CLIPSeg). However, this paper does not compare LangMedSAM with these methods.

- The labels, color schemes and explanatory texts of the charts in the paper are not clear enough, which may affect the readability of experimental results.

- The paper mainly trains and tests the model on 2D slices, and does not verify the model’s spatial consistency and continuity in 3D volumetric data.

**Questions:**

see weakness

---

### Official Review · Reviewer_sN4k · 2025-10-31

**Soundness:** 1
**Presentation:** 1
**Contribution:** 1
**Rating:** 0
**Confidence:** 5

**Summary:**

This paper introduces LangMedSAM, a multi-modal medical image segmentation model that leverages natural language prompts instead of manual bounding boxes. Built upon LiteMedSAM, it supports both text-based and conventional inputs. Trained on 20 public datasets and evaluated on 4 external ones, LangMedSAM demonstrates good generalization and automation potential, providing a scalable solution for large-scale medical image segmentation.

However, the paper’s overall presentation is weak, and the reported results appear rather coarse. There is no statistical analysis of the training data, nor any detailed evaluation of performance across individual datasets. Furthermore, the authors fail to cite the key baseline LiteMedSAM when it is first mentioned, indicating a lack of attention to essential references and academic rigor. Overall, the paper gives the impression that the authors did not approach the work with sufficient seriousness or thoroughness.

**Strengths:**

1. The paper builds up a text-driven medical segment any thing model which I believe is a promising direction for MedSAM.

**Weaknesses:**

1. Poor presentation quality: The paper lacks clarity and organization, making it difficult to follow the methodology and results.
2. Insufficient experimental analysis: No statistical analysis of training data or detailed performance breakdown for individual datasets is provided.
3. No comparison with strong baselines: The study omits comparison with specialized and widely adopted models such as nnUNet, which limits the validity of the claimed performance.
4. Coarse results reporting: Experimental results are presented superficially without in-depth discussion or comparison. Most results are presented in boxplot while due to the limited performance differences, it is hard to see any significant differences in the figure.
5. Missing key citation: Even the baseline LiteMedSAM, which the work builds upon, is not properly cited when first introduced.
6. Lack of academic rigor: The omissions and limited analyses suggest inadequate attention to research completeness and reproducibility.

**Questions:**

To be honest, the paper exhibits too many weaknesses to be considered for acceptance at this stage. Too many critical questions remain unanswered:

1. What is the detailed distribution of the training and evaluation data?
2. How does the model perform compared to specialized models such as nnUNet?
3. Can text prompting effectively enhance data integration or segmentation performance?
...

Overall, the paper is far from meeting the acceptance criteria. It lacks clarity, thorough evaluation, and sufficient experimental analysis. I do not see any questions or revisions that would significantly change my current decision.

---

### Official Review · Reviewer_KAdJ · 2025-11-01

**Soundness:** 3
**Presentation:** 2
**Contribution:** 2
**Rating:** 2
**Confidence:** 5

**Summary:**

This paper introduces LangMedSAM, a model that extends LiteMedSAM to accept natural language inputs for medical image segmentation. The model is trained on a large dataset comprising one million 2D MR or CT slices and is capable of segmentation based on either linguistic descriptions or bounding box prompts. Evaluations on both in-domain and out-of-domain data show promising results, with notable improvements in segmenting thin-walled structures.

**Strengths:**

1. **Enhanced interactivity**. The introduction of natural language inputs to the LiteMedSAM framework is a valuable extension, offering a more flexible and potentially more user-friendly interface for clinical interaction compared to bounding-box-only prompts.
2. **Large-scale evaluation.** The model is evaluated on a substantial dataset. The inclusion of box plots provides a clear visual representation of performance variance across different test cases.
3. **Identified and addressed weakness.** The paper effectively identifies a key limitation of MedSAM and LiteMedSAM, i.e., their difficulty with thin-walled structures (lines 119-120), and provides empirical evidence (Appendix Figure 5) to show that LangMedSAM mitigates this issue.

**Weaknesses:**

1. **Unclear practical value.** The paper motivates the use of natural language by citing the impracticality of bounding-box inputs (lines 058-059). However, this may come at the cost of one of MedSAM's core practical value. MedSAM was assessed in annotation assistance by segmenting *an unseen type* of tumor using bounding-box prompts. The text-image alignment is noted to be redundant in some cases (lines 457-461), which undermines the language model's generalizability in *unseen types*. The practical advantage of LangMedSAM over non-language models needs stronger justification, perhaps through a direct comparison.
2. **An unsubstantiated claim.** The authors claim that algorithms like nnU-Net exhibit "limited generalizability" and perform "suboptimally" on out-of-domain data (lines 040-044), but provide no empirical evidence or citations to support this broad assertion. This claim must be justified to contextualize LangMedSAM's contribution properly.
3. **Questionable out-of-domain generalization.** While the improvement on out-of-domain MR slices (due to better thin-walled myocardium segmentation) is sound, the overall out-of-domain performance is inconsistent. Specifically, LangMedSAM underperforms compared to MedSAM and LiteMedSAM on out-of-domain CT scans (Table 2). Furthermore, in terms of median DSC, Figure 3 shows that LangMedSAM is outperformed by its predecessors for both CT and MR, which raises doubts about its claimed generalizability.

**Questions:**

1. **Evaluation of negative samples.** The training data excluded images with masks smaller than 100 pixels (lines 309-310). Additionally, the DSC computation in Appendix Eq. 12 does not take negative samples (no target tumor/organ in the slice) into account. Are these "negative samples" also excluded from the evaluation? If so, this could artificially inflate performance metrics and misrepresent the model's practical utility, as it may generate false positive segmentations in real-world scenarios where no target is present.
2. **Training stopping criteria.** The stopping criterion is vaguely described as "till convergence" (line 317). Could the authors specify the exact metric used to determine convergence (e.g., loss plateau, DSC on a validation set) and the maximum number of training epochs/steps allowed? This is critical for reproducibility.
3. **Baseline training settings**. The authors state that "The model is trained on a single H200 GPU of 144 GB memory" (line 317) and "all models are assessed under identical conditions" (line 329), but the training configurations for the baseline models are not detailed. Were these models trained from scratch on the same dataset and with the same hardware setup? Clarification is needed for a fair comparison.
4. **Version of BiomedParse**. To ensure reproducibility, please specify which version of BiomedParse (v1 or v2) was used for generating the dataset.
5. **Inconsistent bounding-box performance:** Appendix Table 9 shows that LangMedSAM using bounding-box prompts does not suffer the same performance degradation on MR slices as LiteMedSAM on thin-walled structures (Table 2). This is confusing, given the argument that bounding boxes struggle with such anatomy (lines 119-120, 437-439). Why is LangMedSAM's bounding-box performance an exception to this stated weakness?

---

### Official Review · Reviewer_pMMq · 2025-11-02

**Soundness:** 2
**Presentation:** 1
**Contribution:** 2
**Rating:** 2
**Confidence:** 4

**Summary:**

The paper proposes LangMedSAM, it is a language-adaptive medical image segmentation framework that integrates language prompts into the Segment Anything Model (SAM) paradigm. During this process, the LangMedSAM introduces a lightweight language–vision alignment module that conditions the segmentation process on textual instructions. The model is trained on 20 medical datasets across multiple modalities, and it achieves strong generalization across unseen domains. From this paper, the framework demonstrates that coupling language guidance with lightweight SAM adaptation enables scalable, prompt-based segmentation suitable for clinical and deployment scenarios.

**Strengths:**

1. The paper demonstrates solid engineering that scales SAM-style segmentation efficiently to the medical domain. Training across 20 datasets and achieving good generalization to 4 external benchmarks shows commendable robustness.

2. LangMedSAM maintains competitive performance with only ~700 MB VRAM usage, highlighting its deployment feasibility in clinical or low-resource environments. Such lightweight adaptation is important for translating foundation models to real-world medical applications.

3. The integration of text-based prompts, though conceptually simple, improves accessibility for non-expert users, representing a meaningful step toward more user-friendly medical AI systems.

**Weaknesses:**

1. Limited methodological novelty: The paper mainly integrates existing techniques (connects SAM and a lightweight text encoder) without introducing a fundamentally new learning paradigm or architectural mechanism. The overall framework resembles prior multimodal segmentation systems (e.g., BiomedParse, FLanS [1]), and the contribution lies mostly in engineering refinement rather than conceptual advancement.


[1] Biomedparse: a biomedical foundation model for image parsing of everything everywhere all at once.
[2] FLanS: A Foundation Model for Free-Form Language-based Segmentation in Medical Images

2. Lack of in-depth analysis on language-driven improvements: The paper does not adequately explain why text prompts enhance segmentation performance. The paper evaluates only prompt length, there is no ablation on prompt semantics, robustness to ambiguous or noisy language inputs, or analysis of how linguistic cues influence feature alignment. As a result, the claimed benefit of language adaptivity remains unconvincing.

3. Insufficient differentiation from prior work: Many of the design choices, such as CLIP-based text conditioning and SAM fine-tuning, have been widely explored in earlier works. Without clearer motivation or distinct methodological contributions, the novelty boundary between LangMedSAM and existing frameworks remains weak.

**Questions:**

1. The paper mainly combines SAM with a text encoder — what is the key methodological innovation beyond this integration?
2. How are text prompts generated or standardized during training and evaluation? Are they manually written or programmatically derived?
3. How does the model handle ambiguous or conflicting textual inputs, such as “tumor near left lobe” vs. “mass close to center”?
4. In this work, the authors mentioned "try different text encoders (SAPBERT, PubMedBERT, BERT) and add projection MLPs to match the image side", however, is the language encoder frozen or fine-tuned? If fine-tuned, what data or loss functions guide its alignment with visual features?
5. How does LangMedSAM perform on open-ended or unseen prompts, beyond those directly corresponding to labeled anatomy classes?

---

### Note · Authors · 2025-11-18

I have read and agree with the venue's withdrawal policy on behalf of myself and my co-authors.